# Bayes' Power for Explaining In-Context Learning Generalizations

## Abstract

Traditionally, neural network training has been primarily viewed as an approximation of maximum likelihood estimation (MLE). This interpretation originated in a time when training for multiple epochs on small datasets was common and performance was data bound; but it falls short in the era of large-scale single-epoch trainings ushered in by large self-supervised setups, like language models. In this new setup, performance is compute-bound, but data is readily available. As models became more powerful, in-context learning (ICL), i.e., learning in a single forward-pass based on the context, emerged as one of the dominant paradigms. In this paper, we argue that a more useful interpretation of neural network behavior in this era is as an approximation of the true posterior, as defined by the data-generating process. We demonstrate this interpretations' power for ICL and its usefulness to predict generalizations to previously unseen tasks. We show how models become robust in-context learners by effectively composing knowledge from their training data. We illustrate this with experiments that reveal surprising generalizations, all explicable through the exact posterior. Finally, we show the inherent constraints of the generalization capabilities of posteriors and the limitations of neural networks in approximating these posteriors.

## 1 Introduction

Neural network training is traditionally seen as parameter fitting for maximum likelihood estimation (MLE) or maximum a posteriori (MAP) estimation. One performs multiple epochs of training on a single (small) dataset to get closer to the MLE or to the MAP with regularization.

Nowadays, neural networks are commonly trained in a single-epoch setting though, as the community has moved towards larger or infinite data sources (Brown et al., 2020; Oquab et al., 2024; Hollmann et al., 2023). In these settings, performance is not bounded by available data but compute. The training happens on unseen data sampled from an underlying distribution in each step, thus it is not very useful to think about it as approximating MLE for a dataset which is largely unseen. This paper advocates for understanding neural network training and specifically ICL in this setting as an approximation of the true posterior distribution instead (Müller et al., 2022; Xie et al., 2022). We experimentally show that this perspective predicts the generalization behavior of trained neural networks in ICL settings accurately. Interpreting neural network training as posterior approximation alone cannot fully account for all behaviors of neural networks, given that the architecture, due to its inductive biases, affects how these networks approximate the posterior distribution. As neural networks improve in approximating the posterior with more scale, these biases become less important though if the posterior is representable by the neural network.

In this paper, we train small neural networks with small budgets on infinite artificial data: We can see that the true posterior, shaped by the training data, closely matches the network's behavior in most scenarios. The implications of whether the true data posterior can account for much of a neural network's behavior are significant for the debate over whether large language models (LLMs) are merely pattern matching or reasoning machines. Under the posterior approximation interpretation, LLMs generalize to new data by constructing a posterior that integrates all observed data, contrasting with the concept of stochastic parrots (Bender et al., 2021). The combinatorial power has limitations, though, which we also detail. These limitations pertain to both the quality of the posterior being

subpar for out-of-distribution predictions, as well as the approximation not being tight everywhere, as we detail in this work.

**Contributions** We show the implications of interpreting ICL as a posterior approximation (Müller et al., 2022; Xie et al., 2022) for explaining generalization to unseen data.

- We present examples of unexpected out-of-distribution generalizations on simple analytical tasks.
- We show how these generalizations can be explained and are even expected through the Bayesian interpretation.
- We detail the limits of generalizations in the posterior and the limitations to the posterior approximation quality of neural networks.

## 2 NEURAL NETWORK TRAINING AS POSTERIOR APPROXIMATION

Training a neural network $q_\theta$ on a dataset $D$ is traditionally viewed as finding an approximation to the parameters $\theta$ that maximize the likelihood of the datasets, $\arg\max_\theta q_\theta(D|\theta)$. This is called MLE.

For the nowadays typical single-epoch training (Brown et al., 2020; Hollmann et al., 2023) it is less intuitive to use MLE as an interpretation, however, as large parts of the dataset might not be seen during training. Thus, we are rather working with the data distribution directly. The more natural interpretation which we advocate for is as an approximation to fitting the data generation distribution (Goodfellow et al., 2016, Section 5.5). We approximate

$$\arg\min_\theta \mathbb{E}_{x,y\sim p(X,Y)}[-\log q_\theta(y|x)], \tag{1}$$

where $p(X, Y)$ is the data distribution from which the examples in $D$ are sampled.

The idea for the posterior approximation interpretation, can now simply be found by rewriting Equation 1 as a Kullback-Leibler (KL) divergence minimization with respect to the posterior $p(y|x)$

$$\arg\min_\theta \mathbb{E}_{x,y\sim p(X,Y)}[-\log q_\theta(y|x)] \tag{2}$$

$$= \arg\min_\theta \mathbb{E}_{x\sim p(X)}[\mathbb{E}_{y\sim p(Y|X)}[-\log q_\theta(y|x)]] \tag{3}$$

$$= \arg\min_\theta \mathbb{E}_{x\sim p(X)}[\mathbb{E}_{y\sim p(Y|X)}[\log p(y|x) - \log q_\theta(y|x)]] \quad (\log p(y|x) \text{ is indep. of } \theta) \tag{4}$$

$$= \arg\min_\theta \mathbb{E}_{x\sim p(X)}[D_{\text{KL}}(p(y|x)||q_\theta(y|x))]. \quad (\text{KL divergence definition}) \tag{5}$$

Thus, by minimizing the cross-entropy loss in Equation 1, we implicitly seek a model distribution $q_\theta(y|x)$ that closely approximates the true posterior distribution $p(y|x)$ on data supported by $p(x)$. The exact posterior is recovered with infinite training data, a neural network that can model the distribution $p(y|x)$ and a training that finds the optimum, as previously derived by Müller et al. (2022)[1].

In an ICL setting, our input $x$ is a concatenation of a training set $D_{train} = \{(\boldsymbol{x}_i, y_i)\}_{i\in\{1,\dots,n\}}$ and a query input $\boldsymbol{x}_{query}$ for which we are predicting the output $y = y_{query}$. In this paper, we consider ICL problems where the dataset is sampled by first sampling a latent $l \sim p(l)$ and sampling each example in the dataset based on it $(x, y) \sim p(x, y|l)$. This construction is commonplace in Bayesian modelling and might even be seen in language, where a particular person starts to write a document with a particular intent. Thus, the posterior distribution, also called posterior predictive distribution (PPD) in this setting can now be written as

$$p(y_{query}|\boldsymbol{x}_{query}, D_{train}) = \int p(y|\boldsymbol{x}_{query}, l)p(l|D_{train})dl. \tag{6}$$

In the above representation one can already see the mixture over different latents $l$, that makes the predictions of our models interesting.

---

[1]They focused on in-context learning settings, but the proof can be applied to other input modalities, too.

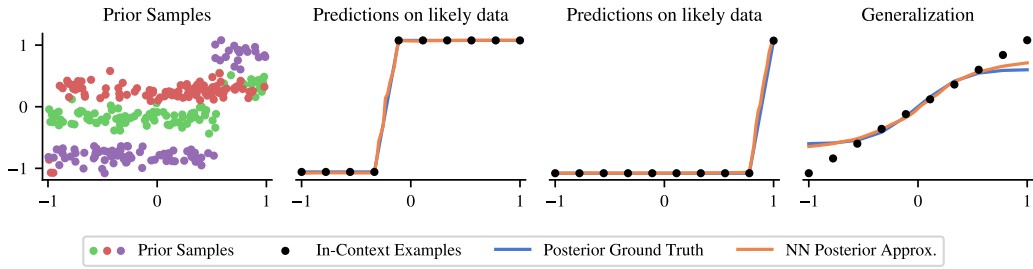

Figure 1: The model is only trained on step functions (left), still it learns to make smooth predictions (right) just like the true posterior for the step function prior.

The optimization from Equation 1 becomes

$$\arg \min_{\theta} \mathbb{E}_{(\boldsymbol{x}_{query}, y_{query}) \cup D_{train} \sim p(\mathcal{D})}[-\log q_{\theta}(y_{query}|\boldsymbol{x}_{query}, D_{train})] \tag{7}$$

and matches $p(y_{query}|\boldsymbol{x}_{query}, D_{train})$ in the limit (Müller et al., 2022).

## 3 IN-CONTEXT LEARNING WITH PRIORS OVER FINITE SETS OF LATENTS

In this work, we look at multiple priors and their posteriors. To allow easily computing the posterior for our priors, we focus on priors with a finite set of latents. That means that the considered priors have latents $l$ that are drawn from a large but finite set $\mathbf{L}$ with equal probability, if not specified otherwise. We built an easily extendable framework to efficiently compute the posterior $p(l|D_{train})$ and the PPD $p(y|\boldsymbol{x}_{query}, D_{train})$ on CPU and GPU to allow further research in understanding in-context learning through the Bayesian lens. In the discrete setting, the PPD, as defined by Equation 6, simplifies to a weighted sum

$$p(y|\boldsymbol{x}_{query}, D_{train}) = \sum_{l \in \mathbf{L}} p(y|\boldsymbol{x}_{query}, l)p(l|D_{train}). \tag{8}$$

For each setup, we define a distribution $p(\boldsymbol{x}, y|l) = p(y|\boldsymbol{x}, l)p(\boldsymbol{x}|l)$ over examples given a latent. If not specified otherwise, $p(\boldsymbol{x}|l) = \mathrm{U}(0, 1)$ is the uniform distribution and we define $p(y|\boldsymbol{x}, l)$ via a deterministic mapping $f : X \to Y$ with output noise $y = \mathcal{N}(f(\boldsymbol{x}), 0.1^2)$.

For all experiments, we use a transformer (Vaswani et al., 2017) adapted for ICL, introduced by Müller et al. (2022) and called a Prior-data Fitted Network (PFN). PFNs are particularly well suited for ICL thanks to their permutation invariance with respect to the set of context/training as well as query examples. For each experiment, we report the mean of the predictions and provide a notebook to reproduce it easily[2]

## 4 GENERALIZATIONS EXPLAINABLE BY POSTERIOR APPROXIMATION

In the following, we will show three examples of interesting generalizations of neural networks when performing ICL, each of which is explainable by the ground truth posterior.

### 4.1 TRAINING ON STEP FUNCTIONS YIELDS SMOOTH PREDICTIONS BUT NOT EVERYTHING REPRESENTABLE

In our first experiment, we show that even when a model was not trained on a smooth function, only on step functions, it becomes a smooth predictor as it approximates the posterior.

---

[2]For each experiment, we performed a grid search for the best final training loss. We searched across 4 and 8 layers, batch sizes 32 and 64, Adam learning rates 0.0001, 0.0003 and 0.001, embedding sizes 128, 256 and 512, as well as 100 000, 200 000 and 400 000 steps. The sizes for our training sets were uniformly sampled between 1 and 100. Code can be found at `https://anon-github.automl.cc/r/BayesGeneralizations-19B2`.

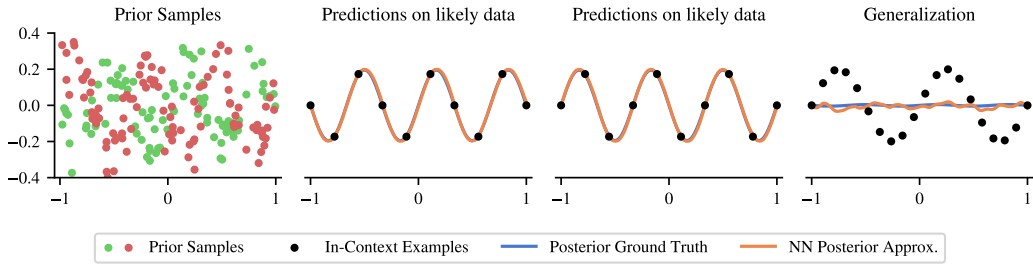

Figure 2: Training a model on sine curves of a single amplitude, frequency and different offsets (left), the model does not only learn to model these curves (center), but also models the posterior for a sine that has a wavelength of 2, instead of 3. The posterior is flat, as the model is very uncertain about the offset $\Delta x$ of this curve.

During training, we sample step functions, see Figure 1 (left), that start on different heights $\Delta y$, have different step positions $\Delta x$ and step sizes $h$. We can define the set of functions in our latent set as

$$
\mathbf{L}^{(1)} = \left\{ f^{(1)}_{\Delta x, \Delta y, h} : \begin{aligned} \Delta x &\in \{-1., -0.98, \dots, 1\}, \\ \Delta y &\in \{-1., -0.98, \dots, 1\}, \\ h &\in \{0, 0.02, \dots, 2\} \end{aligned} \right\},
\tag{9}
$$

$$
\text{where } f^{(1)}_{\Delta x, \Delta y, h}(x) = \begin{cases} \Delta y & \text{if } x < \Delta x \\ \Delta y + h & \text{else.} \end{cases}
\tag{10}
$$

The predictions in Figure 1 (right) are smooth, as there are multiple step functions that might have produced the line, and the PPD now averages all of these step functions as in Equation 6. Thus, even though our model was trained solely on non-smooth step functions, its predictions are (approximately) smooth as they are an average of many step functions.

## 4.2 Training on Sine Curves Can Yield Flat Line Predictions

Similar to the above experiment, we observe that training on sine curves with different offsets can generalize to an unseen function, a flat line, if the data comes from a sine curve that has a different frequency from the sine curves observed during training.

Our prior is defined as follows

$$
\mathbf{L}^{(2)} = \left\{ f^{(2)}_{\Delta x} : \Delta x \in \{0, 2\pi/100 \dots, 2\pi\} \right\},
\tag{11}
$$

$$
\text{where } f^{(2)}_{\Delta x}(x) = 0.2\sin(3\pi x + \Delta x)).
\tag{12}
$$

In Figure 2 (right) we show that the model closely matches the ground truth posterior for a sine curve $sin(2\pi x)$ that is outside the training distribution. This again is explained by a plethora of latents being averaged in the predictions of the model.

## 4.3 Training on Sloped Lines and Flat Sines Teaches Predicting Sloped Sines

We now show that models can even mix two different classes of functions. We show that models trained on sine curves at different heights and on lines at different heights and slopes, generalize to sine curves that have a slope.

This was previously thought not to be possible, as elaborated by Yadlowsky et al. (2023) for example. It turns out that this is not only possible, but to be expected according to the posterior approximation interpretation.

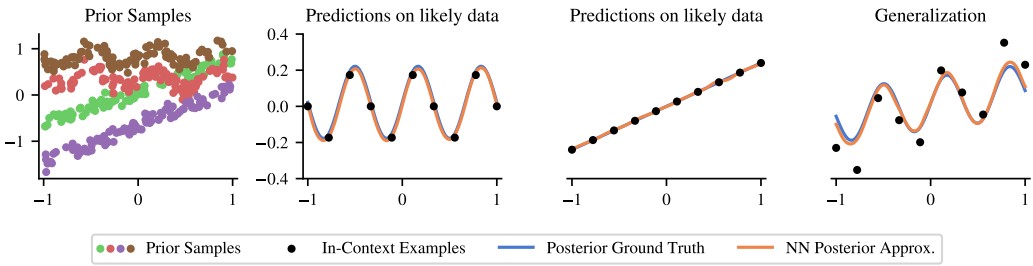

Figure 3: We train a model on two distinct classes of functions, sines and sloped lines, only (left). It not only learns fit both function classes well (center), but also learns to model slightly sloped sines, when prompted with a data from a sloped sine.

Our latents are a uniform mixture of the previous prior $\mathbf{L}^{(2)}$, and a prior over sloped lines

$$\mathbf{L}^{(3)} = \left\{ f_{\Delta x,m}^{(3)} : \begin{matrix} \Delta y \in \{-1., -0.98, \ldots, 1\}, \\ m \in \{-1., -0.98, \ldots, 1\}, \end{matrix} \right\} \tag{13}$$

$$\text{where } f_{\Delta x,m}^{(3)}(x) = mx + \Delta y \tag{14}$$

In Figure 3, we can see that the generalization to a slightly sloped sine curve is possible, and that it even is in agreement with the true PPD. Here, the PPD is a linear mixture of sine functions and lines and can thus represent a wave function that has a slope.

For this prior, we performed an ablation that considers different training times to see whether models tend to converge to the Bayes optimal prediction as training goes on. In Figure 4 we see that the approximation of the posterior becomes better as we perform more training steps, but seems to suffer diminishing returns as we increase the number of training steps.

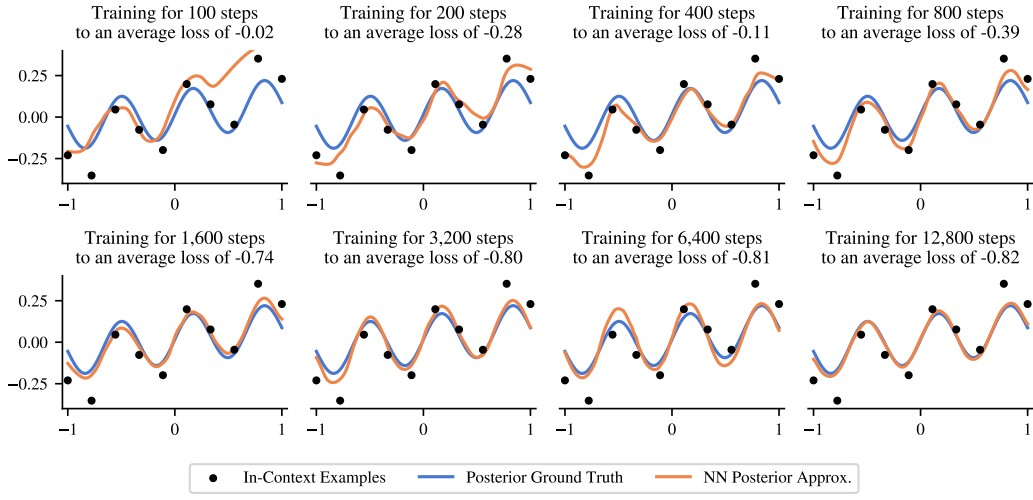

Figure 4: We see that the approximations of the true posterior become better with more training steps and a lower cross-entropy loss, like we expect for a powerful model as outlined in Section 2. The losses are negative, as we are in a regression setting, where the density can be above 1.

## 5 LIMITATIONS OF THE POSTERIOR

While we show interesting generalizations that are explainable via posterior approximation above, in this section we will focus on the limitations of generalizations supported by the posterior itself. That is, in which scenarios can't we expect the posterior to yield useful and intuitive predictions.

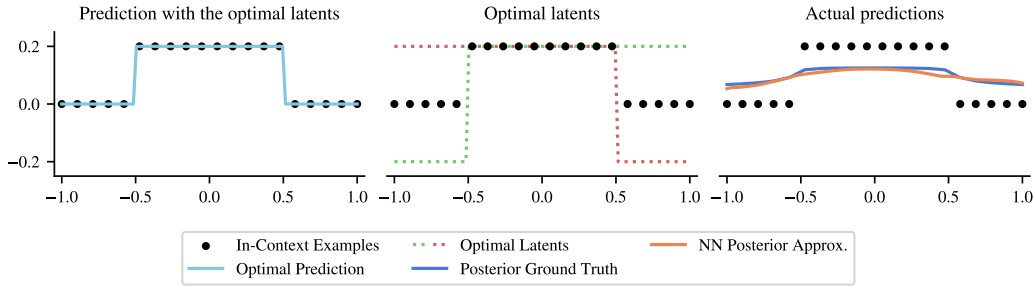

Figure 5: While the model could make the optimal prediction (left) using a posterior mixing just the two latents in the center, it predicts differently as the latents in the center have a low likelihood to have generated the data.

## 5.1 BEING REPRESENTABLE IS NECESSARY BUT NOT SUFFICIENT

A combination of latents that perfectly fits the data does not guarantee the posterior uses this or an equivalent combination. We demonstrate this by extending the step prior $\mathbf{L}^{(1)}$ from Section 4.1 to include negative steps, i.e., $h \in \{-1, -.98, \ldots, 1\}$.

The prior consists of simple step functions, but combining two yields a mean modeling a step up followed by a step down (Figure 5, left). This can be achieved by many posterior distributions, such as one assigning $50\%$ probability to each of the latents in Figure 5 (middle). Thus, it could theoretically fit the data perfectly. Figure 5 (right) illustrates the actual posterior and its approximation, though. The prediction is much flatter than it should be. This is because the latents that would yield the right distribution do not get enough probability mass in the posterior, as they predict parts of the data very poorly (in our example the left- and right-side). Thus, while there is a combination of latents that optimally approximates the function, the posterior does not choose that combination; this does not even change when providing more data (as this would also lead to more data points being modelled poorly by each of the latents that could be mixed to fit the function optimally).

## 5.2 BAYESIAN MODELS WITH MISSPECIFIED PRIORS BECOME EXPONENTIALLY WORSE

Bayesian ensembles tend to concentrate on a smaller subset of the prior as one conditions on more data. While this is beneficial if the to-be-fitted function has support in the prior, it is detrimental if it does not. We say a function has support in the prior, if there is a latent that shares the mean prediction with the function. In general, certain functions lie outside the support of any prior distribution. Even Gaussian processes lack support for specific functions, such as step functions.

In cases where the function does not have support in the prior, the posterior will still concentrate. It will concentrate on a close-by latent in terms of KL-divergence (Burt et al., 2020), but a necessarily

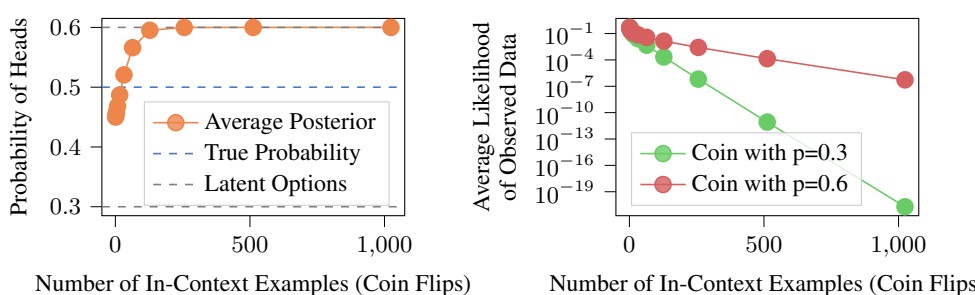

Figure 6: We evaluate a Bayesian model's performance in estimating coin bias, using latent options $p = 0.3$ and $p = 0.6$. The posterior deteriorates with more data as the likelihood for $p = 0.6$ dominates. Results are averaged over all potential coin toss outcomes.

wrong latent. And as the likelihood for a latent has a multiplicative form ($\prod_i p(x_i, y_i)$), it becomes exponentially more confident in the wrong latent as more data is provided.

In Figure 6, we demonstrate this effect analytically on a simple coin flip prior. Here, we use two discrete latents: either the coin has a head probability of $p = 0.3$ or $p = 0.6$. The coin flips we condition on are actually fair ($p = 0.5$), though. On the left, we show that while the posterior on average is close to the actual true probability after observing only a few coin tosses, it diverges towards the closer latent ($p = 0.6$) as more coin tosses are observed. On the right, we see that this is due to the likelihood of $p = 0.6$ growing exponentially compared to $p = 0.3$.

We can further see this effect for more interesting priors, too. In Figure 11 of the appendix, we show that when we feed more data to our prior that combines sine curves and lines (Section 4.3), we see the same effect of converging to a wrong solution while starting from a good prediction. Finally, in Figure 12 of the appendix we show this effect even happens for a Gaussian process, when modelling a step function. The Gaussian process becomes increasingly worse at modelling the step as more ground truth data is provided.

# 6 LIMITATIONS TO THE POSTERIOR APPROXIMATION INTERPRETATION

While the posterior approximation interpretation is useful to predict generalizations in many cases, it has limitations which we showcase experimentally in the following section. We first look at the unreliable behavior of neural networks when we are outside of the support set of the prior, then we look at cases where we are inside the support but with low probability. Finally, we discuss model limitations that might hamper the ability of models to approximate the posterior.

## 6.1 APPROXIMATING AN UNKNOWN DISTRIBUTION, YIELDS UNKNOWN OUTCOMES

The intuition of posterior approximation breaks down when the input data $x$ has no support ($p(x) = 0$) in the training distribution (extrapolation). The posterior $p(y|x) = p(x, y)/p(x)$ is simply not defined in this setting. In this section we show that neural networks tend to behave unpredictably in this setting.

In Figure 7 we provide evidence that neural networks are less reliable in extrapolation settings. Here we train a standard MLP on a balanced binary classification problem: classify 0s and 1s, i.e., the examples all have the form $(x = 0, y = 0)$ or $(x = 1, y = 1)$. While the models are all able to predict well on the support set $\{0, 1\}$, in between these values, where we are outside the support, each model behaves differently. As soon as we add a bit of Gaussian noise (standard deviation of $0.1$) to the inputs during training, though, the behavior of our models becomes much more predictable on the $(0, 1)$ interval[3].

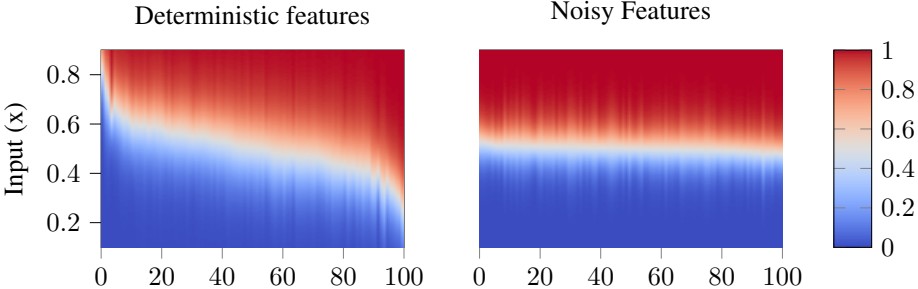

Figure 7: Predictions on the $[0, 1]$ interval of 100 MLPs (x-axis) trained with different seeds, sorted by their prediction at $0.5$. The noiseless training data (right) results in less reliable outcomes, with models exhibiting greater variations in their predictions based on their seed.

---

[3]All models are 3-layer MLPs with ReLU activation and hidden size 64. They were trained with Adam Kingma & Ba (2015), a learning rate of 0.001, and a batch size of 1024. The experiment can be reproduced using the notebook `https://anon-github.automl.cc/r/BayesGeneralizations-19B2/Tiny_MLP_Generalization.ipynb`.

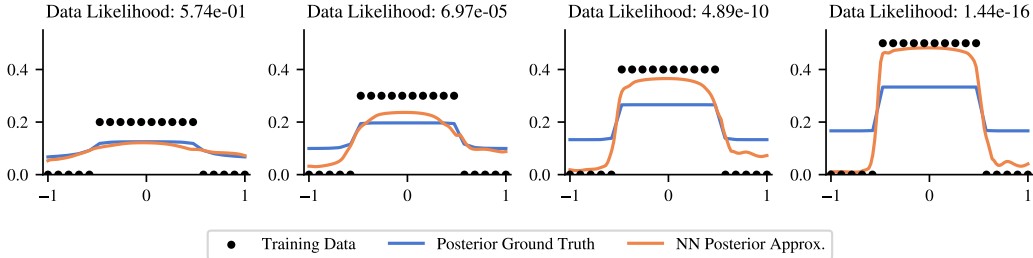

Figure 8: This illustrative example of PFN behavior with decreasing data likelihood shows that the model aligns with the posterior for well-supported datasets but reverts to shortcut predictions with less likely data. Above each plot we provide the data likelihood according to the prior described in Section 5.1.

We posit that extrapolation ($p(x) = 0$) is typically unreliable in neural networks and should thus be avoided, e.g., by augmenting training data to ensure a well-defined posterior. The definition of support is only precise up to invariances and equivariances of the neural network though, e.g., a translation-invariant model will behave predictably when translating inputs, even outside the support set.

## 6.2 THE THRESHOLD OF SUPPORT

While for the exact posterior the support set is well defined, for the neural network-based approximation there is no apparent difference between a likelihood very close to zero and exactly zero, both are typically not in the training data. It might have included a very similar one, though.

We can see, however, that prediction quality decreases with the data likelihood. We showcase this in Figure 8 for the prior from Section 5.1, where this phenomenon is easy to see. For the high likelihood data, the approximation still is close to the true posterior, but as we move the lines further apart, thus making our dataset less likely, the neural network falls back to a nearest-neighbor-based prediction.

While in the limit, we expect to have correct predictions wherever we have support, we see that in practice the predictions deteriorate before that.

## 6.3 ARCHITECTURAL LIMITATIONS

Neural networks are constrained by the limited complexity of functions they can model due to finite computational resources, thereby unable to accurately represent certain posterior distributions. Consequently, predictions do not converge to the true posterior despite minimizing the loss function.

We illustrate a recognized limitation in architectures akin to PFNs, which we employ: encoder-only transformers lacking positional embeddings fail to count repeated inputs (Barbero et al., 2024; Yehudai et al., 2024). This is due to the nature of self-attention involving weighted averages in a permutation-invariant fashion. Thus, a posterior that involves counting the number of identical inputs will not be well approximated by our models.

To demonstrate, we construct a simplistic prior focused solely on estimating the probability of a coin landing heads, devoid of any additional features. Coins are sampled each with a distinct head probability $p \in \{0.01, 0.02, \ldots, 0.99\}$ for different datasets, and data samples are generated by flipping the coin with the outcome assigned as the target $y$. The setup is illustrated at the top of Figure 9 (left). The posterior corresponding to this prior must incorporate the decrease in uncertainty as additional samples are observed. Solely heads samples were inputted, as depicted in Figure 9 (left, bottom), to estimate the probability of obtaining heads on a subsequent coin flip. Despite the influx of identical samples, the neural network's prediction remains unchanged, as shown in Figure 9 (right), irrespective of the actual variation in the posterior.

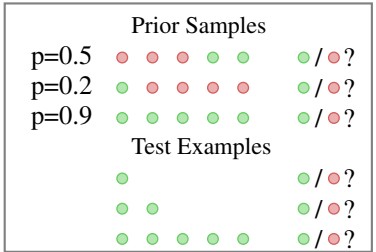 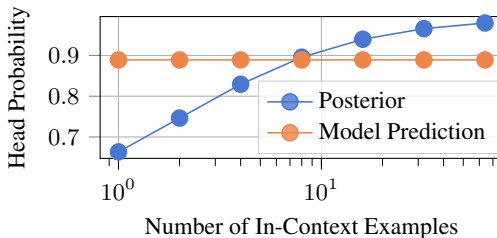

Figure 9: On the left (top), we outline our prior, sampling a probability $p$ for heads (green) and generating samples by coin flips. At test time (left, bottom), we condition on varying counts of coins displaying heads. On the right, we can see that the transformer-based PFN model is not able to approximate the posterior, as it would need to count the number of examples, which all are identical, in the context set.

The inductive biases inherent in neural network architectures manifest less critically in specific phenomena, such as the consistent errors observed across different models in our experiments on modeling sloped sines. These models consistently overestimate the sine magnitude relative to the true posterior as the slope increases. The predictions of the eight best models are presented in Figure 10 in the appendix.

Investigating the precise modeling constraints and inductive biases of contemporary neural networks constitutes an active research domain (Bhattamishra et al., 2020; Weiss et al., 2021), potentially augmenting the probabilistic insights obtained from the posterior.

# 7    RELATED WORK

While interpreting neural network training as posterior approximation in ICL was established in previous work (Xie et al., 2022; Müller et al., 2022), this study investigates its explanatory power in single-epoch ICL setups and identifies its limitations.

Yadlowsky et al. (2023) previously discussed the generalization behavior of ICL models. They found that transformers do not have particular inductive biases making them especially strong on out-of-distribution ICL. What we have added to their analyses is that data alone, defining a posterior, enables generalization to new input-output mappings.

Finally, there is a line of work using the interpretation of neural network training as posterior approximation, namely PFNs (Müller et al., 2022; Hollmann et al., 2023; Müller et al., 2023; Rakotoarison et al., 2024; Adriaensen et al., 2023), which are trained on data sampled from a prior as an approximation to the posterior. PFNs are a form of amortized inference, where the PPD (mapping from training data and test example to test output) is learned directly. Thus, they heavily rely on the posterior approximation interpretation.

# 8    CONCLUSION & FUTURE WORK

In this paper, we demonstrate that transformers trained for in-context learning (ICL) can achieve notable out-of-distribution generalizations, that are unlike a nearest-neighbor matching, and involve the composition of training examples. We found that these generalizations can be explained by the posterior that the transformer learns to approximate implicitly in many cases. We did also find, though, that these generalizations exhibit clear limitations and may not function intuitively in all scenarios. If the data-generating function does not have support in the prior distribution, model predictions tend to converge to incorrect solutions. Moreover, when conditioning on data that lies outside the support of the prior, neural network behavior becomes less reliable and predictable.

Open future work are the following points. i) Develop a practical definition of *support* in neural network training, considering that most data within the support is not sampled during training. ii) Understand what types of priors models prefer to approximate. This is interesting because the data sampled from the prior typically does not identify the prior. iii) Finally, we would like to learn if

this interpretation can be useful for understanding more about the behavior of language models in general sequence settings.

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

## A APPENDIX

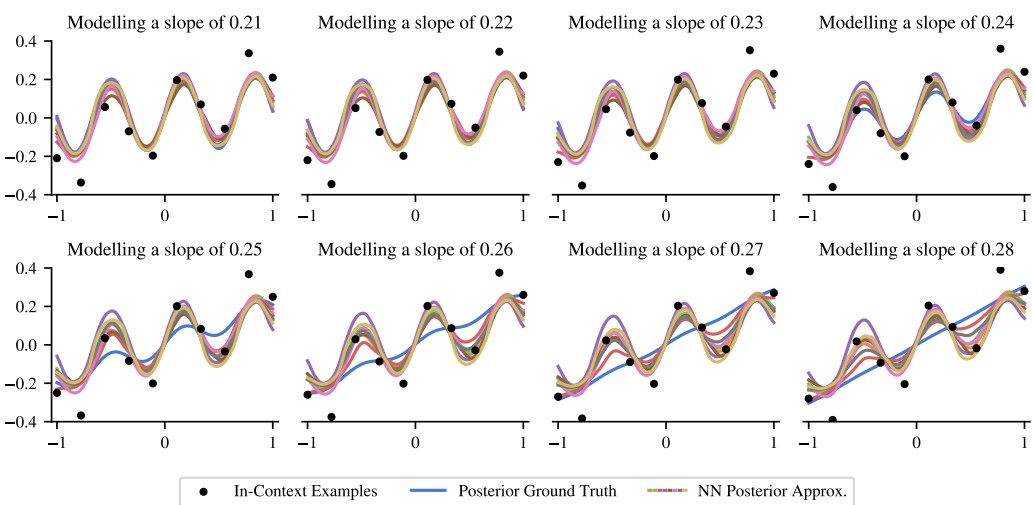

Figure 10: We show the predictions of the 8 strongest models from our grid search. We see that all models tend to make similar mistakes on this task, which is likely due to a bias in the architecture.

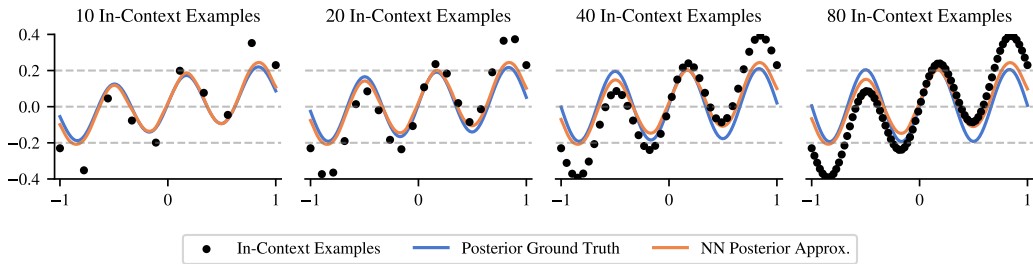

Figure 11: The prior combining sines and lines introduced in Section 4.3 deteriorates in accuracy as more data points are conditioned upon, as detailed in Section 5.2. Notably, the neural network approximation makes an erroneous approximation here and still predicts a sloped sine. This model was trained on dataset with up to 100 examples, thus this is not due to length generalization.

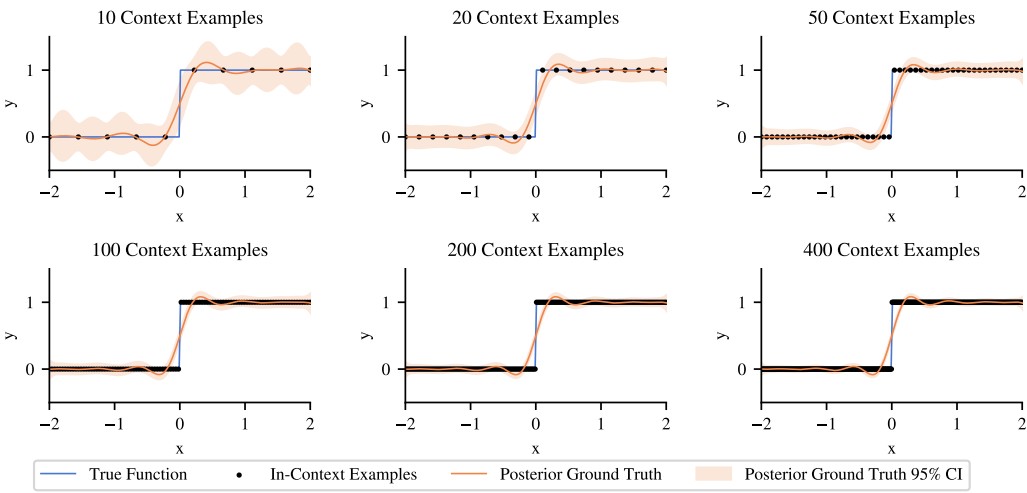

Figure 12: We show a simple Gaussian Process (GP) conditioned on 10 to 400 linearly spaced context points of a step function. The GP, with an RBF kernel (lengthscale 0.4, outputscale 1.0), a constant mean function, and Gaussian noise ($\sigma = 0.1$), mostly models the step within its 95% confidence interval at first but increasingly becomes over-confident in an over smooth function as more points are added. Reproduce this experiment with our notebook `https://anon-github.automl.cc/r/BayesGeneralizations-19B2/GP_fitting_a_step.ipynb`.

