# OpenReview forum: "Bayes' Power for Explaining In-Context Learning Generalizations"
_ICLR.cc/2025/Conference — Submitted to ICLR 2025_

### Official Review · Reviewer_JQVe · 2024-10-30

**Soundness:** 3
**Presentation:** 2
**Contribution:** 2
**Rating:** 5
**Confidence:** 4

**Summary:**

This paper studies transformers that are adapted for ICL in the sense of transformers trained a la Muller et al., 2022. This involves a meta-training type of objective that leads to Eqn (7). The majority of the hard results center on trying different classes of latents $L$ and deterministic mappings $f: X \to Y$ which go onto define the meta-training datasets in Eqn (7). Transformers trained in this way are seen to generalize to unseen tasks and this can be explained because the transformers trained in this way are trying to approximate the Bayes posterior predictive density (PPD).

**Strengths:**

* This work leans on the strengths of Muller et al., 2022 which showed that Transformers can be trained using an ICL-objective to give us a good approximation of the Bayes PPD. Because there is a rich literature of the generalization behavior of the Bayes PPD, we can explain the ICL behavior of transformers trained for ICL by appealing to this classic theory.
* There's a good discussion of the limitations of their conjecture

**Weaknesses:**

* I found the second and third paragraphs in Section 2 really out of place and to contain confusing terminology. For instance $p(y|x)$ is repeatedly referred to as the true posterior distribution. Is this a typo?
* The description of the transformer adapted for ICL feels inadequate. There is a disconnect between presenting Eqn (7) and what the transformer has to do with it. At present, the paper is not intelligible without first reading Muller et al., 2022
* I did not find the related work very adequate. There are in total three paragraphs in Section 7, the last one talks about PFNs. The first one is one sentence describin Xie et al 2022 and Muller et al 2022 as "interpreting neural network training as posterior approximation in ICL". The second paragraph mentions one paper Yadlowsky et al. 2023 as discussing the "generalization behavior of ICL models". What is an ICL model?

**Questions:**

* Section 2: Would anything be lost if you removed paragraphs 2 and 3, "For the nowadays..." and "The idea for the posterior approximation..."?
* Section 3: "We built an easily extendable framework to efficiently compute the posterior $p(l|D_{train})$" Where do you do this...? I thought your framework is for computing the PPD via Eqn (7) which actually bypasses the posterior entirely.
* Section 3: "we define $p(y|x,l)$ via a deterministic mapping $f: X \to Y$ with output noise $y=N(f(x),0.1^2)$". Where did the dependence on $l$ go?

---

### Official Review · Reviewer_Zx1r · 2024-11-02

**Soundness:** 3
**Presentation:** 4
**Contribution:** 2
**Rating:** 5
**Confidence:** 4

**Summary:**

Recent work has argued that ICL can be interpreted as Bayesian inference, where the set of examples are taken from a randomly sampled function, and the goal is to produce the posterior predictive (PPD) for the test example. This paper shows how the PPD interpretation can help to understand generalization behavior in ICL. A series of experiments train a modified transformer on some prior distribution of functions and test its out-of-distribution generalization, comparing its output to both the generating function and the exact posterior.

**Strengths:**

The paper advances an interesting ideal-observer perspective on ICL, that we can understand generalization just by looking at the posterior implied by the data (without reference to the NN architecture).

More specifically, it points out that out of distribution testing leads to posteriors that are mixtures of functions in the prior.

**Weaknesses:**

The question in practice is not how well the network approximates the PPD, but how well it learns the true generating function. The paper gives examples of successful generalization where the posterior (and the network) can be close to the generating function even when it's outside the training distribution (fig 3), and unsuccessful generalization where the posterior and the network are far from the generating function (fig 2). But then there are cases where the posterior is far from the generating function yet the network is close to it, because the network does not approximate the posterior well (fig 8). In the end one has to question how useful the paper's core PPD thesis is for explaining ICL generalization.

Sec 5.2 shows that, as the number of examples grows from zero, the posterior starts at the prior mean, moves toward true function, but then converges to the KL-closest member of the prior support. So it may pass near the true function but only transiently. I worry Examples like fig 3 are cherry-picked for a nice intermediate point on this trajectory (i.e., a sweet spot in number of examples). The paper would be stronger if it focused on the full trajectory.

Fig 2: wavelength should be wave number

**Questions:**

It would help to include a few sentences describing the PFN.

One of the paper's main points is the posterior is a weighted mixture of functions in the prior, where the weights are the posterior on the latent l. It would be useful to see the posterior on l, especially in fig 3 and 5.

The asymptotic posteriors as in figs 11-12 are helpful in connection with my comment above about the full learning trajectory. It would be useful to see the same thing for fig 5.

The idea that generalization performance might be nonmonotonic (i.e., best for some intermediate number of examples) is interesting. Can this happen for language tasks? That seems like it would be a significant result, assuming it hasn't been demonstrated before.

Fig 10: how is strongest defined?

---

### Official Review · Reviewer_dSKU · 2024-11-04

**Soundness:** 3
**Presentation:** 3
**Contribution:** 3
**Rating:** 5
**Confidence:** 3

**Summary:**

This paper studies ICL as doing bayesian inference by approximating the true posterior defined by the data-generating process. The authors propose function classes that make this Bayesian interpretation of ICL evident, with experiments showcasing surprising generalizations that can be fully explained through the exact posterior. Additionally, the authors highlight the inherent limitations of posterior-based generalization, as well as the architectural constraints of neural networks.

**Strengths:**

1. The experiments demonstrating the interpretative power of ICL are clear and compelling, with ideas presented in a straightforward and illustrative manner.

2. The authors also examine inherent constraints that lead to common failures, such as analyzing the X shift (out of support).

**Weaknesses:**

1. Previous literature has raised questions regarding the Bayesian nature of ICL. For example, Raventós et al. showed that transformers pre-trained on data with low task diversity struggle to learn new tasks and identified a threshold beyond which ICL emerges. Numerous studies suggest that phase transitions occur with respect to both the diversity of the training data (the cardinality of $L$ here) and the context sequence length (the number of context tokens used both during training and inference). Additionally, some work has observed a simplicity bias, where neural networks tend to “prioritize” learning simpler patterns first. It appears that the authors have not sufficiently addressed these factors in relation to the Bayesian interpretation of ICL.

2. This paper only focuses on the finite discrete prior case. There is extensive literature studying the case of continuous priors (e.g. regression case). For example, in bayesian linear regression one might draw the weights $w\sim N(0,\sigma^2$). It is unclear how this study’s insights extend to such cases -- while it's possible to interpret the $w$'s seen in pre-training as the $L$ set, it does seem quite unnatural and contrived to interpret the unseen $w$'s as some mixture of the seen $w$'s. Additionally, the model architecture requires the data to be exchangeable, which complicates generalization to Markov settings or more complex language tasks.

3. Extending from the common Bayesian linear regression case, it is also a well-known challenge in the field that problem dimensionality correlates closely with the difficulty of training a model to “learn to learn” (ICL). For instance, even Bayesian linear regression becomes challenging when $w$ has dimension on the order of hundreds. In more complex tasks like language modeling, where embedding dimensions often range from hundreds to thousands, it is unclear how useful these insights are relative to the impact of engineering choices and training techniques.


Allan Ravent´os, Mansheej Paul, Feng Chen, and Surya Ganguli. Pretraining task diversity and the
emergence of non-bayesian in-context learning for regression, 2023. URL https://arxiv.
org/abs/2306.15063.

**Questions:**

1. Just making sure, the posterior ground truth you are plotting is the posterior predictive mean? i.e. $E_{\text{true posterior predictive}} [y_{query}\mid x_{query}, D]$. And these were analytically solved?

---

### Official Review · Reviewer_mZnP · 2024-11-04

**Soundness:** 2
**Presentation:** 3
**Contribution:** 2
**Rating:** 3
**Confidence:** 4

**Summary:**

The authors point out that a common informal way of thinking about neural network training, namely that training is approximately searching for the maximum likelihood estimate of the parameter given the data, is not appropriate in a setting like large language model training where models are trained in a single epoch. They advocate instead focusing on approximation of the “posterior” by which they mean the true distribution of data from which the “infinite” stream of examples is drawn.

In order to explore this hypothesis they train a series of transformers on in-context learning (ICL) tasks, where (x,y) pairs are presented in context y = f_l(x) + epsilon for some latent l fixed in each context but varying between contexts and normally distributed noise epsilon. This is a common testbed for understanding the nature of ICL in recent literature.

They show that transformers trained in such a setting, where the f_l are either step functions or sin functions, appear to learn to predict according to a kind of Bayesian posterior predictive distribution, which marginalises out the latent l. This is a hypothesis for ICL that has been explored extensively in recent years, in works including Xie et al “An explanation of in-context learning as implicit Bayesian inference” and Raventos et al “Pretraining task diversity and the emergence of non-Bayesian in-context learning for regression” NeurIPS 2023 and the overall situation is (as far as I know) still unclear.

**Strengths:**

- The study of step functions as a simple class of functions for in-context learning is novel, as far as I know.

**Weaknesses:**

- I am confused by the use of the term “posterior” in this paper. For someone with a background in Bayesian statistics the posterior is a distribution over model parameters, and I would refer to the p(x,y) in this paper as the true distribution and p(y|x) as the true conditional distribution, rather than the posterior. I think the authors are following the terminology of Xie et al, in which marginalising out the latent vector produces a kind of posterior predictive distribution (thinking of the latents as the parameters for a kind of “in-context model”) but this does not mean p(x,y) is the posterior.
- The paper does not situate itself well enough in the literature, including Raventos et al as already mentioned, and also Garg et al “What Can Transformers Learn In-Context? A Case Study of Simple Function Classes”.

**Questions:**

While the central claim of the paper is interesting, I already believe in this conclusion based on Raventos et al and do not currently see what additional evidence this paper provides.

Things that might change my views:

- A more thorough comparison to related work including Garg et al and Raventos et al, and clarifying the additional contributions made by this paper. What do step functions tell me that linear regression does not?
- In this paper it is somewhat implicit that the population loss for the training objective does have (6) as its global minima, something that is e.g. close to proven in Raventos et al. Is it possible to argue for this theoretically? Arguably then this setting of step functions is more “minimal” than linear regression.

Some more technical notes:

- Could you more carefully define the “posterior ground truth” and “NN posterior approx” in Fig 1-5? I think “NN posterior approx” just means the output of the transformer and “posterior ground truth” just means the true function, in which case it is unnecessarily confusing terminology.
- Also the authors use “prior samples” to mean samples from the training distribution. While it might be appropriate to think of the transformer as doing in-context Bayesian inference with the training samples as the “prior” and outputs as the PPD, this is an empirical hypothesis to be proven, not something that should be labelling plots a priori (in my opinion, and if I am understanding the setting correctly).

---

### Official Review · Reviewer_LxmC · 2024-11-10

**Soundness:** 2
**Presentation:** 4
**Contribution:** 1
**Rating:** 3
**Confidence:** 5

**Summary:**

Going beyond the usual MLE interpretation, this work takes a Bayesian interpretation of in-context learning. The authors highlight that autoregressive probabilities in a sequence model are posterior predictives. They demonstrate how sequence models extrapolate to unseen latent factors (e.g., predicting smooth outputs from step functions). The authors note  sequence models may generalize poorly when the data is not supported in the model’s prior.

**Strengths:**

The paper offers examples that illustrate how sequence models generate posterior predictions on latent factors unseen at training time. I thought the experiments were thoughtfully designed to highlight the basic Bayesian interpretation. While widely known in other contexts, the paper's discussion on the limitations of the posterior approximation interpretation is useful and adds color to the authors' narrative. I found the discussions on how PFN-style attention cannot capture repetitive patterns is illuminating.

Overall, the authors' illustrations are informative and interesting and suggest immediate research questions.

**Weaknesses:**

The paper provides interesting toy examples that suggest natural research questions to explore, but does not address any of them in depth. I truly appreciate the visualizations and the discussion in the submission. However, the paper currently reads as if it is a well-written report on toy simulations at the beginning of a research project, or a lecture note surveying well-known facts about sequence models and open research questions. It is unclear to me what the authors' main contributions are, if it is to be peer-reviewed as a conference proceeding.

Having said this, I see ample opportunities for the submission to grow into a fully fleshed paper.

First, I suggest authors study the relationship between the capacity of the model and its extrapolation behavior in-depth. A sufficiently flexible model can exhibit any arbitrary smoothing behavior, and the inductive bias built into the model will govern its extrapolation behavior. It is thus unclear to me why the generalization/smoothing behavior is explained by the Bayesian posterior approximation view alone. I see a couple of different ways to proceed. On the empirical side, one can build scaling law on how model capacity impacts ICL extrapolation behavior could be a valuable contribution to the literature. On the other hand, the authors could also analyze extrapolation behavior theoretically, and provide insights as to how transformers introduce particular behavior on unseen latents.

A Bayesian model provides a way to combine the observed data points to infer the latent structure (l in the paper's notation). To my understanding, the current paper does not provide substantive insights on how sequence models (e.g., PFNs) perform this synthesis process in an implicit manner. The phenomenological discussion on the posterior predictive distributions are true tautologically but in my opinion, does not add much to our understanding of the robustness of ICL (or lack thereof). Furthermore, as the authors also observe, Bayesian models with misspecified priors can lead to incorrect predictions, and increasing data can worsen performance under such conditions. It is obvious that sequence models will suffer the same fate, but a detailed understanding of how their behavior under misspecification will be valuable.

Another research question of interest is a careful study of different sequence modeling architectures. The authors solely focus on PFNs but as the authors also note, this particular modification to the attention masking mechanism suffers limitations. My understanding is that there are alternative masking approaches such as Nguyen et al. or the more recent work by Ye et al., as well as recent works such as Sun et al. that propose new state space modeling architectures (see below for proper citations). The impacts of different modifications to attention or SSMs appears poorly understood in the literature, warranting further analysis.

Finally, the external validity of the observations in the paper are not clear and seem to apply specifically to the particular experimental setting studied. Even with resources available in typical academic labs, for ICLR I would expect a careful study of scaling behavior up to 100M-1B of parameters across different environments.

As a minor feedback, I see some gaps in the authors' discussion of the literature. As the paper notes, the interpretation of autoregressive models as learning posterior predictives is well-established. More broadly, the meta-learning literature has long taken this view of sequence modeling.

In addition to Xie et al. and Muller et al. cited in the submission, several authors have contributed to this growing literature, e.g., see Jeon et al. or Nguyen and Grover below. Going beyond this observation, a number of research groups have explicitly drawn connections between autoregressive generation and posterior inference, e.g., see Zhang et al. Ye et al., and Falck et al..

- Jeon, H. J., Lee, J. D., Lei, Q., & Van Roy, B. (2024). An information-theoretic analysis of in-context learning. *arXiv preprint arXiv:2401.15530*.
- Nguyen, T., & Grover, A. (2022). Transformer neural processes: Uncertainty-aware meta learning via sequence modeling. *arXiv preprint arXiv:2207.04179*.
- Zhang, L., McCoy, R. T., Sumers, T. R., Zhu, J. Q., & Griffiths, T. L. (2023). Deep de Finetti: Recovering topic distributions from large language models. *arXiv preprint arXiv:2312.14226*.
- Ye, N., Yang, H., Siah, A., & Namkoong, H. (2024). Pre-training and in-context learning is bayesian inference a la de finetti. *arXiv preprint arXiv:2408.03307*.
- Falck, F., Wang, Z., & Holmes, C. (2024). Is In-Context Learning in Large Language Models Bayesian? A Martingale Perspective. *arXiv preprint arXiv:2406.00793*.
- Sun, Y., Li, X., Dalal, K., Hsu, C., Koyejo, S., Guestrin, C., ... & Chen, X. (2023). Learning to (learn at test time). arXiv preprint arXiv:2310.13807.

**Questions:**

N/A

---

### Meta-Review · Area_Chair_vD9w · 2024-12-20

**Metareview:**

This paper proposes a Bayesian interpretation of in-context learning (ICL) in large language models, arguing that neural network training can be viewed as an approximation of the true posterior defined by the data-generating process, rather than maximum likelihood estimation. The reviewers praise the paper's interesting and well-designed experiments, which demonstrate the interpretative power of ICL and highlight the limitations of posterior-based generalization, but criticize the paper for lacking depth and clarity in its contributions, with some reviewers feeling that the paper reads like a report on toy simulations or a lecture note surveying well-known facts, and that the authors do not sufficiently address related work or provide substantive insights into how sequence models perform implicit Bayesian inference. Overall, all reviewers lean towards rejection and the authors have not provided any rebuttal. We would still like to encourage the authors to resubmit an improved version of the paper in the future.

**Additional Comments On Reviewer Discussion:**

no rebuttal submitted

---

### Decision · Program_Chairs · 2025-01-22

Reject